# Current Clinical Landscape and Global Potential of Bacteriophage Therapy

**DOI:** 10.3390/v15041020

**Published:** 2023-04-21

**Authors:** Nicole Marie Hitchcock, Danielle Devequi Gomes Nunes, Job Shiach, Katharine Valeria Saraiva Hodel, Josiane Dantas Viana Barbosa, Leticia Alencar Pereira Rodrigues, Brahm Seymour Coler, Milena Botelho Pereira Soares, Roberto Badaró

**Affiliations:** 1School of Medicine, University of Missouri, Columbia, MO 65201, USA; 2SENAI Institute of Innovation (ISI) in Health Advanced Systems, University Center SENAI/CIMATEC, Salvador 41650-010, BA, Brazil; 3Gonçalo Moniz Institute, FIOCRUZ, Salvador 40291-710, BA, Brazil; 4School of Medicine, University of California San Diego, San Diego, CA 92093, USA; 5Elson S. Floyd College of Medicine, Washington State University, Spokane, WA 99202, USA

**Keywords:** bacteriophages, antimicrobial resistance, phage therapy, review

## Abstract

In response to the global spread of antimicrobial resistance, there is an increased demand for novel and innovative antimicrobials. Bacteriophages have been known for their potential clinical utility in lysing bacteria for almost a century. Social pressures and the concomitant introduction of antibiotics in the mid-1900s hindered the widespread adoption of these naturally occurring bactericides. Recently, however, phage therapy has re-emerged as a promising strategy for combatting antimicrobial resistance. A unique mechanism of action and cost-effective production promotes phages as an ideal solution for addressing antibiotic-resistant bacterial infections, particularly in lower- and middle-income countries. As the number of phage-related research labs worldwide continues to grow, it will be increasingly important to encourage the expansion of well-developed clinical trials, the standardization of the production and storage of phage cocktails, and the advancement of international collaboration. In this review, we discuss the history, benefits, and limitations of bacteriophage research and its current role in the setting of addressing antimicrobial resistance with a specific focus on active clinical trials and case reports of phage therapy administration.

## 1. Introduction

Bacteriophages, or phages, are viruses that infect and replicate within various bacteria and do not carry the potential to infect eukaryotic cells [1]. There are an estimated 10^31^ phages; their near-ubiquitous distribution partly facilitates the astonishing abundance of these infectious agents. Bacteriophages exist wherever there are bacteria and are therefore found in nearly every environmental matrix as well as in animal and human guts [2,3,4,5]. The viruses that comprise the realm of phages have each evolved to rely on specific bacterial hosts for survival. The molecular machinery of these hosts facilitates the proliferation of the phage, in some cases sufficiently disrupting metabolic processes, leading to bacterial lysis and death [5]. This ability of phages to exploit bacteria provides an exciting tool for combating pathogenic bacteria. The therapeutic potential of bacteriophages has therefore garnered worldwide attention in the last few years, offering promising applications in combatting difficult-to-treat bacterial infections when antibiotics fail.

Bacteriophages were discovered in the early 20th century, and their therapeutic utility for treating certain infections was acknowledged shortly thereafter [6,7,8]. The logistics of phage research presented considerably greater practical challenges than antibiotic development, and the original studies evaluating phage therapy in clinical settings yielded statistically insignificant or questionable findings. As favorable and scalable outcomes were achieved from antibiotic research, antibiotics swiftly monopolized the market and gained investment by large pharmaceutical companies, tipping the scales away from phage development. Several political and social issues also contributed to a diminished interest in phage research. World War II and the Cold War fostered a global landscape in which novel discoveries were not readily shared between the United States (U.S.), the Soviet Union (USSR), and their respective allied nations. As a result, while antibiotic research and production boomed across most high-income countries (HICs), the USSR continued to prioritize bacteriophage therapy [9]. Antibiotic discovery, research, and production throughout the 20th century made antibiotics the cornerstone of bacterial disease management while phage research and therapeutic development dwindled into relative obscurity.

Antibiotics have served as a cornerstone of modern medicine for most of the past century. Bacterial infections, however, continue to cause substantial loss of life and health across the world, disproportionately impacting lower-middle-income countries (LMICs) which are nations with a gross national income per capita of less than USD 4125 [10,11]. The widespread and excessive utilization of antibiotics in medical and agricultural settings has contributed to rapidly evolving antimicrobial resistance (AMR) to many of the therapeutics developed over the past eighty years [12]. Growing resistance to this method of combating infection has challenged the clinical utility of even the most efficacious antibiotics. Those species of bacteria posing the gravest threat to the efficacy of antibiotics today include *Enterococcus faecium*, *Staphylococcus aureus*, *Klebsiella pneumoniae*, *Acinetobacter baumannii*, *Pseudomonas aeruginosa*, and *Enterobacter faecium*, collectively known as the ESKAPE pathogens [13]. These pathogens are known to subvert the antimicrobial action of the most widely available antibiotics. With the escalating threat of AMR, bacteriophage therapy has returned to the forefront as a potential tool to mitigate a rapidly worsening global health crisis.

In this review, we discuss the current state of bacteriophage research—phage discovery, classification, and production. We describe the potential therapeutic application of phages, particularly regarding their use in addressing antibiotic-resistant infections. We also review challenges of developing new drugs to combat AMR and emphasize the importance of international collaboration in phage research and industrial production.

## 2. Bacterial Antimicrobial Resistance

### 2.1. Overview of AMR

AMR is characterized by the ability of bacteria to develop or acquire the ability to render antimicrobial agents less effective in their bacteriostatic or bactericidal properties [14]. There are many ways in which bacteria can develop resistance against antibiotics; such mechanisms include cellular efflux pumps, mutations to drug targets, and antibiotic-inactivating enzymes such as beta-lactamases [15,16]. In most cases, bacterial antimicrobial-resistance determinants have the capacity for horizontal transfer, causing the spread of factors that confer resistance within a bacterial species population as well as between different species of bacteria. The horizontal transfer of AMR genes is predominantly accomplished by conjugation, transduction, or natural transformation; each process involves the transfer of transposable elements, which are mobile DNA sequences that can change their position within a genome and can provide the code for the various mechanisms that confer antibiotic resistance [14]. Emerging resistance among bacterial pathogens leads to infectious diseases which are difficult to treat with the existing arsenal of antibiotics, which can spread across the globe, and which inevitably increase the risk of severe illness and death in those affected [17,18].

The ESKAPE bacteria are among the most infectious, deadly, and costly pathogens responsible for the increasing prevalence of AMR infections. The World Health Organization (WHO) has declared the development of alternative treatments for ESKAPE pathogens a global priority [19]. Characterization of antimicrobial resistance includes delineation into three groups: multidrug-resistant (MDR), extensively drug-resistant (XDR), and pan-drug-resistant bacteria (PDR). MDR refers to bacteria resistant to at least one antibiotic in at least three antibiotic categories; XDR is resistant to one agent in all but two categories; and PDR is resistant to all antibiotics [20]. Even those clinical bacterial isolates which are not PDR remain exceedingly dangerous and can lead to an increased risk of death, extended hospital stays, and prolonged patient recovery, especially in resource-poor regions such as LMICs where accessibility to preferred second- or third-line antibiotic substitutions can be limited due to high costs or inadequate drug delivery pipelines [11].

### 2.2. Emergence and Progression of AMR

Bacteria have faced environmental pressures for 3.5 billion years, driving their evolution to select mechanisms of resistance and survival long before the industrial-scale manufacturing of modern antibiotics [21]. However, the production and distribution of large quantities of antibiotics has undoubtedly increased the selection pressure that eradicates bacteria susceptible to antibiotic regimens and promotes the survival and proliferation of bacterial strains better equipped to counter antimicrobial compounds [22,23].

The emergence of antibiotic-resistance genes and, consequently, the rise of antimicrobial-resistant bacteria have been clearly associated with the extensive use and misuse of antibiotics [15]. Antibiotics are the most frequently prescribed medications in the U.S.; annually, an estimated 833 prescriptions for antibiotics are written per 1000 persons [22]. Within the healthcare system, the misuse of antibiotics via inappropriate prescription practices, insufficient patient education contributing to non-adherence to prescribed antimicrobial regimens, and limited pharmaceutical regulatory oversight all contribute to the rising rates of AMR [24]. Subinhibitory concentrations of antibiotics, in particular, lead to physiological changes in bacteria, which may lead to increased resistance and changes in virulence [25]. Antibiotics are not only used in human and veterinary medicine, but also in agricultural and aquacultural settings where they are utilized at an industrial scale to protect valuable livestock, additives for plant farming, and for food decontamination [26]. Bacteria with AMR capacities evolve in these non-medical settings and can be transferred to humans via food consumption, direct contact with livestock, and environmental spread in the form of runoff water and human sewage [24].

Rising AMR in LMICs is often a manifestation of individual or national poverty and has been linked to weaker local and national health systems as well as disrupted supply chains [24,26]. In India and Uganda for example, the limited availability of diagnostics and therapeutics can compel physicians to prescribe antibiotics based on availability rather than indication [11]. Other drivers of AMR in LMICs include cost-prohibitive pricing for antibiotics that are then unaffordable and inaccessible for many local residents, as well as limited government expenditures on subsidized healthcare, a high prevalence of over-the-counter access to antibiotics, minimal training on antibiotic stewardship, inappropriate antibiotic storage leading to drug degradation and substandard dosing, and the utilization of counterfeit or off-brand alternatives that may be more affordable but less effective [24,26]. There are a wide range of other underlying factors that contribute to the selective pressures that produce AMR; however, one of the most concerning may be the trend of a disproportionately increased use of novel or last-resort antibiotics in numerous LMICs [11]. These various applications, among many others, have facilitated the widespread presence of antibiotics in the environment. Many of these compounds are now commonly reported in wastewater, animals in the human food chain, landfills, and industrial waste [27]. This increasingly expanding utilization of antibiotics has exerted a substantial selective pressure on a wide range of bacterial species. Consequently, the transmission of numerous different species of MDR bacteria and, accordingly, cases of MDR infections in humans, are now increasingly being reported not only in controlled environments, such as healthcare facilities, but also within communities at large. Once sufficiently capable of destroying an array of infectious pathogens, antibiotics are becoming increasingly obsolete against a rising tide of these highly resistant deadly organisms [28].

### 2.3. Epidemiology of AMR Infections

In 2021, the WHO declared that AMR is among humanity’s top 10 global public health threats [29]. It is calculated that, in the United States alone, more than 2.8 million people contract an infection resistant to traditional antibiotics annually, contributing to more than 35,000 deaths, with similar numbers reported in Europe [30,31]. In 2019, AMR was directly responsible for an estimated 1.27 million deaths worldwide—surpassing the deaths attributed to malaria and HIV/AIDS that year [32]. The COVID-19 pandemic caused by the severe acute respiratory syndrome coronavirus 2 (SARS-CoV-2) further exacerbated the existing global antimicrobial resistance crisis via significant increases in antibiotic prescription, the widespread use of disinfectant agents, and an increased risk of the nosocomial transmission of AMR pathogens among patients with longer intensive care unit (ICU) stays [33,34]. A 2014 review on AMR found that by 2050 the global mortality rate from AMR infections will rise to greater than 10 million deaths annually, with a substantially disproportionate impact on LMICs across Asia and Africa compared to the U.S. and Europe [10,35,36,37,38]. While the COVID-19 pandemic provides a bold reminder of the potential impact of infectious diseases on global health, some public health specialists propose that AMR may pose an even more sinister, “silent pandemic” [39].

### 2.4. Economics of AMR

Substantial economic impacts of AMR are brought on by the increasing costs of preventing and mitigating MDR bacterial spread as well as treating the infections caused by these microorganisms. By 2050, the annual cost of addressing AMR globally is projected to be approximately USD 300 billion to USD 1 trillion [37,38]. In the European Union, MDR bacterial infections contribute to additional healthcare costs and lost productivity that total at least EUR 1.5 billion per year (approximately USD 1.64 billion) [39]. An increased length of hospital stays, complicated and extended antibiotic regimens, and the volume of tests run on patients with MDR bacterial infections contribute significantly to the rising costs [40]. Additionally, the financial burden can be influenced by the species of bacteria responsible and the extent of the infection’s clinical manifestations [41]. Similar trends are seen regarding carbapenem resistance. In 2020, Zhen et al. reported estimated differences in total hospital costs of USD 14,251, USD 4605, and USD 7277 per patient with carbapenem-resistant *K. pneumoniae*, *P. aeruginosa*, and *A. baumannii*, respectively. The length of stay was also higher for all three bacterial types when comparing carbapenem-resistant versus carbapenem-sensitive isolates, with the most notable increase of 15.8 days identified among infections with *A. baumannii* [42]. In addition to increasing mortality rates compared to sensitive strains, multi-drug-resistant *P. aeruginosa* has also been shown to contribute to an average adjusted total cost of USD 22,370 per patient [43].

## 3. Antibiotics

Since the end of the 20th century, there has been relatively little headway in novel antibiotic discovery and production. Remarkably, there are currently no antibiotics at the clinical trials stage of development that aim to enhance or expand antimicrobial coverage against the ESKAPE pathogens that are not derivatives of existing antibiotic classes [18].

Novel antibiotic discovery has proven difficult, as clinically useful compounds must be efficacious and present good toxicity profiles. Most of the recently approved antibacterial agents are not representative of novel classes but are instead derivatives of existing classes, developed primarily from compounds with established antibacterial action that can be used as templates for further drug development [44,45]. One problem with this approach is that these emerging analogs carry substantially higher risk and rate of drug resistance as existing bacterial defense mechanisms quickly adapt through selective pressures to provide augmented antibiotic resistance. According to the World Health Organization’s 2020 review, there were 11 new and approved antibacterial drugs by 2017; however, only two met WHO’s “innovation criteria” and constituted novel chemical classes [46].

## 4. Bacteriophages

### 4.1. Phage Overview

Bacteriophages were first discovered in the early 1900s by two bacteriologists: Frederick William Twort and Félix d’Hérelle [47]. The recognition of their utility in combating infections did not take long after their discovery—and d’Hérelle even described their use in treating *Vibrio cholerae* outbreaks as early as the 1920s—when phage prophylaxis against Asiatic cholera was introduced to numerous villages in India via direct, individual administration as well as public water supply treatments. The results were impressive: the outbreak was halted within 48 h as opposed to the almost month-long waiting period associated with typical prophylactic approaches at the time [8]. It did not take long for phage preparations to reach other nations, including Brazil, Egypt, Italy, and the United States, where they were utilized to treat several bacterial diseases, including dysentery and typhoid, among others [48].

Phages are incredibly diverse with genome sizes ranging from 2435 bp to over 500,000 bp in the case of aptly named mega-phages [4,49,50]. Similar to other viruses, phages contain DNA or RNA genomes that are single or double-stranded. When a phage infects a bacterium, its DNA classically follows one of two reproduction cycles: lytic or lysogenic. Following a lytic cycle, a phage utilizes the bacteria’s molecular machinery to produce its progeny. Upon rapid and extensive proliferation, the new phages cause the bacteria to lyse, allowing for further phage–host interaction. In contrast, phages in the lysogenic reproduction model infect the host bacterium and incorporate its DNA directly into the bacterial genome, thereby passing genetic material onto each host’s offspring as the bacterium engages in replicative cycles. The incorporated viral genome is known as a “prophage”, notably distinct from viral DNA integrated into eukaryotes, known as “provirus”. These phages are called temperate phages and may remain in the dormant lysogenic cycle replicating its DNA within the bacterial chromosome until specific genes are triggered by appropriate conditions (i.e., ultraviolet radiation, temperature, oxidative stress, etc.) to activate. This activation promotes prophage induction, by which the phage DNA exits the chromosome to proliferate en masse, thereby lysing the host bacterium [1]. In the context of potential therapeutic applications, lytic phages are predominantly selected for clinical studies. Lysogenic phages are posited to be more likely to transfer virulent genes from host bacteria with AMR properties to other, less-virulent bacteria. This can occur by way of transduction (one of the primary methods of the horizontal transfer of AMR genes discussed earlier in this review) or by lysogenic conversion in which non-essential prophage genes (e.g., Cholera toxin, Shiga toxin, and Vibrio toxin) are transferred into the bacteria, altering the phenotype and lending to the development of resistance mechanisms [51]. Lysogenic phages, therefore, pose a significant risk of exacerbating, rather than ameliorating, antimicrobial resistance [14]. While lysogenic bacteriophages have the potential to create more virulent infections in humans and animals, their lytic counterparts have the distinct capability to be utilized to combat AMR infections as the lytic process effectively destroys the bacterial host [1,14,52].

### 4.2. Classification, Taxonomy, and Genomic Diversity

The organization of phages has evolved with the advent of genetic sequencing technology [49]. Historical classification of bacteriophages began with David Bradley in the 1960–70s and included morphologic types, such as tailed, filamentous, and icosahedral phages, further separated based on a genetic material organization (e.g., single-stranded DNA or RNA) [53]. Phage classification in recent years continued to utilize morphology but was additionally augmented by other characteristics such as genome, synteny, proteome, and phylogeny [54,55]. In 2023, the Bacterial Viruses Subcommittee of the International Committee on Taxonomy of Viruses (ICTV) modified the taxonomic classification of phages by abolishing the morphology-based families *Myoviridae*, *Podoviridae,* and *Siphoviridae* as well as the order *Caudovirales,* which became the class *Caudoviricetes.* This change was made following multiple independent assessments that determined that morphologic classification did not accurately reflect their shared evolutionary histories [56,57].

More than 44,000 completed bacteriophage genomes are available through the National Center for Biotechnology Information (NCBI) GenBank, an incredible increase from 9722 in 2019 [55,58]. Many bacteriophages discovered thus far are characterized by double-stranded DNA (dsDNA) and tails, primarily comprising a class of Caudoviricetes phages isolated by the Science Education Alliance Phage Hunters program (SEA-Phages) [59]. However, the diversity of bacteriophage families is hypothesized to expand substantially over the next several years as more laboratories become involved in isolation and characterization and new phages are identified from among diverse environments. Half of the complete phage genomes currently published on NCBI utilize only seven bacterial genera as hosts (i.e., *Mycobacterium*, *Streptococcus*, *Escherichia*, *Pseudomonas*, *Gordonia*, *Lactococcus*, and *Salmonella*). This indicates that there likely exists a vast array of novel bacteriophages and phage families to discover in this field.

Bacteriophages that utilize different species of bacteria as hosts vary significantly in their genetic composition. However, even among bacteriophages that utilize the same genus of host bacteria, there exists impressive genetic diversity [58]. Studies employing pairwise comparisons indicate minimal, if any, genetic similarity, even at the amino-acid sequence level between phages [60]. While the full implication of genomic differences amongst phages has not yet been elucidated, genomic differences appear to result in variations among phage properties. This subsequently impacts propagation methods, namely lytic versus lysogenic reproductive strategies, as well as receptor binding proteins that aid viral entry into a bacterium. Furthermore, genetic differences lead to variability in polymerase specificity in these phages, generating narrow specificities and host ranges. One example of this is the bacteriophage RAD2, which encodes a specific depolymerase that enables it to target and degrade a hypervirulent capsule employed by strains of *Klebsiella pneumoniae* [61]. Many phages possess these essential polymerase-encoding genes that inadvertently aid in the receptor–receptor–binding protein interaction and facilitate the infection of a specific bacteria [61].

Structurally, bacteriophages consist of a capsid head in which the genetic content of the virus is stored. The mechanism of transfer of this genetic material into bacterial cells is made possible by what can be described as a “syringe-like” tail which binds to specific receptors on the host cell [1] (Figure 1). The interaction between bacteriophage tail receptor binding protein (RBP) and bacterial cell receptors plays a vital role in the host range of bacteriophages. Phages known to bind to one specific receptor are called monovalent phages, while those that bind to multiple are referred to as polyvalent bacteriophages [62,63]. Intuitively, these differences in receptor binding affinity result in phages with either a narrow host range (i.e., the capability to target only one bacterial strain) or a broad host range (i.e., the ability to target both different bacterial strains within the same type of bacteria as well as bacterial strains from distinct species) [64,65]. However, host switching by modulating RBPs is possible in some bacteriophages. For example, the phage BPP-1 which targets the bacterium *Bordetella* can modulate its RBP-encoding gene, *mtd*, via reverse transcriptase [66,67]. This propensity for RBP modulation is also seen in the phage T4, which is originally isolated against *Escherichia coli*, but with modulation of certain hypervariable domains, increases host range inclusion of *Yersinia pseudotuberculosis*. Even apart from direct modulation, some phages encode more than one RBP at baseline and can change the specific receptor binding protein being expressed (e.g., Mu, P1 phages against *Enterobacteria*) [68].

Host range is further defined by a phage’s capacity to evade or suppress a given host’s intracellular defenses as well as by the ability of phage progeny to lyse and exit the cell. Bacterial cell lysis partially depends on specific protein variations that facilitate endolysin-mediated degradation of the cell wall [68]. Notably, however, various studies have illustrated antagonistic pleiotropy among phages, demonstrating that as host range increases (by way of any of the aforementioned phage characteristics), there appears to be an inverse trend in phage virulence [68,69].

### 4.3. Discovery and Isolation of Bacteriophages

The majority of the phages listed in NCBI were discovered through the SEA-Phages program at the Howard Hughes Medical Institute. SEA-Phages has a comprehensive guide for discovering and characterizing bacteriophages that can be easily adapted to discovering bacteriophages across numerous host bacteria and sample types [59]. Although specific protocols for bacteriophage discovery differ based on laboratory preferences, there are generally four steps: isolation, purification, amplification, and characterization. The traditional method for discovering bacteriophages involves isolating a bacteriophage from the environment using a bacterial host strain of interest (Figure 2). Environmental sources for phage discovery are predominantly water samples from rivers, lakes, hospital waste, and sewage treatment plants.

Two main methods exist for the isolation of a bacteriophage from the environment: direct inoculation and enrichment [59,70]. The direct inoculation method allows for more phages to be isolated so long as they can survive in the lab conditions and infect the strain of bacteria of interest. However, because the concentration of bacteriophage may be relatively low within an environmental sample, the chances of isolating a phage are greater with the enrichment strategy.

The protocols for phage therapeutic development are easily adaptable to a wide range of laboratory settings, even those with limited resources. Often produced as a dry powder formulation, phages can also be stored and transported without refrigeration. Moreover, emerging findings from clinical studies have shown that, in contrast to antibiotics, which often require sequential dosing to reach pharmacokinetic and pharmacodynamic parameters, even small single doses of phages are often sufficient due to the inherent expansion of viral particles following cellular infection [71]. Such characteristics of phage production and administration make them an ideal candidate for large-scale industrial manufacturing and broad application, particularly in LMICs [72].

## 5. Phage Therapy

### 5.1. Current Landscape

The first known successful clinical use of intravenous bacteriophage therapy in the U.S. occurred in March 2016 at the University of California San Diego, where a phage preparation was used to treat a severe multi-antibiotic-resistant *Acinetobacter baumannii* infection. This case helped reignite interest in bacteriophage therapy throughout the nation, resulting in formal public presentations, a book, and various other efforts aimed at increasing public interest in the potential of phage therapeutics [73,74,75]. In August 2021, the FDA released transcripts from a meeting discussing the “Science and Regulation of Bacteriophage Therapy”, in which the logistics of phage application in clinical settings were discussed at length amongst leading scientists and medical practitioners [76]. Given the extensive impact of antibiotic-resistant bacterial infections, bacteriophage therapy has been granted emergency use authorization (EUA) by the U.S. Food and Drug Administration (FDA). This EUA allows for the compassionate use of this therapy on a case-by-case basis. Compassionate use of investigational therapies, otherwise known as expanded access, is when patients identified as having no other therapeutic options (as determined by a physician) may receive products that regulatory agencies have not fully approved. These compassionate care cases occur within a controlled clinical environment. However, they do not occur within the setting of clinical trials and are therefore evaluated through the case series of individual patients rather than larger cohorts of selected populations.

Commonly, the optimal candidates for the compassionate use of bacteriophages are patients who suffer from recurrent and chronic infections, such as patients with cystic fibrosis who are predisposed to frequent pulmonary infections and corresponding antibiotic use, which increases with chronicity of the underlying disease. The extensive antibiotic resistance commonly seen in these patients presents the circumstances in which phage therapy administration via compassionate care approval may be the only remaining option for treatment. A phage therapy study conducted in 2005 by The Center of Phage Therapy Unit (PTU) at the Ludwik Hirszfeld Institute of Immunology and Experimental Therapy in Poland evaluated 153 patients with different MDR infections. The study evaluated both the efficacy and the safety of phage therapy and the results suggested that a significant percentage of patients with chronic bacterial infections were able to tolerate phage therapy, yielding good clinical outcomes [77]. The Antibacterial Resistance Leadership Group (ARLG) reviewed 63 single patient investigational new drug (SPIND) cases in which phage therapy was administered to patients with severe or life-threatening illness, finding that 51 of these cases resulted in favorable outcomes (i.e., clinical cure, clinical resolution, clinical improvement, recovered, a decline in semiquantitative bacterial burden) [78]. Similar promising results were also yielded from a retrospective analysis of 153 patients with a wide array of infections who were treated with phage therapeutics between January 2008 and December 2010; the data indicated that phage therapy yields promising clinical outcomes among a significant cohort of patients with chronic bacterial infections unresponsive to antibiotics [77]. The ARLG concluded that phage formulations are generally well-tolerated, which aligns with the previous FDA designation of certain natural phages as “Generally Recognized as Safe” for use as food additives to prevent bacterial contamination. However, there is still a wide heterogeneity in protocols, disease indications, quality assurance (QA), quality controls (QC), and a lack of placebo controls, making it difficult to determine the efficacy of phage therapy as a whole [78].

Several other single-patient phage administrations have demonstrated encouraging clinical outcomes, indicating that this therapy is generally a successful adjunct for difficult-to-treat infections [76]. For instance, Aslam et al. (2020) described the use of intravenous phage therapy in ten patients with MDR infections at the University of California San Diego. Their report showed successful outcomes in seven of the ten cases, and no safety concerns were mentioned. Of the infections treated, causal bacteria were diverse, including *A. baumannii*, *P. aeruginosa*, *S. aureus*, and *E. coli* [73]. Little et al. (2022) demonstrated promising results by utilizing bacteriophages against a disseminated cutaneous *Mycobacterium chelonae* infection, for which the patient had previously been unsuccessfully treated with a regimen involving ten different antibiotics [79]. The patient reported flushing after each administration of the phage preparation but did not experience adverse effects of the therapy; he experienced a significant improvement in his condition within the first two weeks of therapy [79]. Promising endeavors in bacteriophage therapy have also included utilization in treating MDR infections in transplant recipients and secondary bacterial infections in COVID-19 patients, among others [80,81]. A recent review of phage therapy described an amalgamation of early case reports where phage therapeutics or combination therapy with antibiotics and phages were provided to a total of 70 patients with varying types of infections, including cutaneous infection (3%), prostatitis (9%), burns (9%), endocarditis (9%), intra-abdominal infections (9%), disseminated infections (9%), urinary tract infections (9%), osteomyelitis (9%), implanted devices (30%), pulmonary infections (17%), and other site infections (9%) [82].

Phages have been administered in various regimens, including once daily, twice daily, once every six to eight hours, or over continuous infusion; no standardized schedule is currently recommended by the ARLG Phage Taskforce [78]. Similarly, the route of administration and duration of treatment varies based on the disease and a physician’s clinical judgment, as is the case with many treatment modalities. For example, chronic relapsing urinary tract infections have been treated with twice daily oral phage preparation and twice daily bladder irrigation, treated over 12 weeks [83]. Cases of *P. aeruginosa* ventilator-associated pneumonia, on the other hand, have been successfully treated with intravenous and nebulized phage preparation twice daily for seven days [84]. While there remains room for clinical judgment in these facets of phage therapy, strict guidelines are followed to limit the level of endotoxin (a dangerous byproduct of phage purification) present in phage formulations for human use. A maximum limit of <5 endotoxin units/kg body weight/h has been set by the FDA. Dosage for phage preparations is communicated in plaque-forming units (PFUs), and while dosage varies amongst case reports, concentrations of 10^8^ PFU/mL have been generally determined to be effective phage densities for clinical phage preparations [57,85].

Results of compassionate use cases accessible to the public indicate promising outcomes. However, the implications from such cases must be considered alongside the limitations of how they are conducted; in each case, phage therapy has been co-administered with the gold standard: antibiotics. These cases are further complicated by variations in dosage, route of phage administration, and clinical setting in which the compassionate use cases are executed. Furthermore, although the FDA generally requires clinicians to demonstrate the efficacy of phages before administering the therapy, many cases do not specify the endpoints to measure efficacy or the phage target. This becomes imperative in heterogeneous infections, in which a phage cocktail may be effective against one species of bacteria while ultimately proving unsuccessful in eradicating all pathogenic bacteria [86]. Additionally, there may be an underlying tendency for academic and scientific journals to publish case reports of successful phage therapy applications rather than cases where the introduction of phage proved unsuccessful.

Furthermore, limited standardization among the patients selected for these case studies permits substantial variation in each individual’s severity of chronic illness and superimposed bacterial infection. Recent efforts to standardize these studies are best exemplified by the Hirszfeld Institute of Immunology and Experimental Therapy (HIIET) in Poland which adapted and established new regulations to appropriately meet their growing interest in phage therapy research. The institution enacted new standards that guide the development of phage preparations, including required approval by a bioethics committee, stipulations that trials be conducted by an individual physician or by an approved health center, close adherence to the rules imposed for therapeutic experiments as detailed in the Act on the Medical Profession (1982), and strict compliance with the guiding principles outlined under the World Medical Association in the Declaration of Helsinki. The therapeutic protocol employed at the HIIET is periodically modified and improved, with approval granted by the standing bioethics committee [77].

Although case reports are valuable in highlighting what may be a promising and significant clinical utility of bacteriophages, it is challenging to control for potential confounders. Further evaluation by way of clinical trials will be necessary to ascertain statistical significance. Regardless, it is notable that in many cases where phage therapy has been introduced as an adjunct therapy to antibiotics, disease course, and severity of infection diminish [78]. Consider the report of phage therapy used to treat recurrent UTI caused by ESBL-producing *K. pneumoniae*. The patient had been unsuccessfully treated with varied regimens of meropenem seven times and only achieved infection resolution after phage therapy initiation [83]. Antibiotic-phage synergy experiments performed in vitro also support the possibility of “phage adjuration”, by which the effect of antibiotics on bacteria is “restored” even for bacterial species with resistance genes [87]. Phage-antibiotic studies have also revealed that combined treatment not only impacts antibiotic resistance, but can also contribute to decreased phage-resistance in select bacterial hosts [88,89]. However, this is not a universal finding, as phage-antibiotic combination therapy has also been shown to select for phage resistance [90]. These studies are an important step in determining valuable phage-antibiotic combinations which may be applicable in the clinical setting, as well as which combinations should be avoided.

### 5.2. Phage Therapy Clinical Trials

A critical transition point for any scientific endeavor is from the observed and anecdotal to the replicable and quantifiable. Bacteriophage therapy still appears to be reaching an inflection point in its progression from qualitative to quantitative, as seen by many recent clinical trials. A total of seven phage therapy clinical trials conducted between the years 2000–2015 are listed on clinicaltrials.gov. In contrast, 18 clinical trials were initiated in 2022 alone. As of March 2023, a total of 45 clinical trials are listed on clinicaltrials.gov and an additional completed phase I/II clinical trial from 2015 examining phage therapy for *P. aeruginosa* burn wound infections was listed on the European analogue website (clinicaltrialsregister.eu, accessed on 18 April 2023). Of these, 25 are phase I or phase II clinical trials, with 14 combined phase I and phase II. There are five phase III and no phase IV trials. The remaining studies include expanded use, observational, or preclinical studies [91]. Of the published phage clinical trials, 11 are not yet recruiting patients, while 13 are active, and are already completed (Table 1).

Sponsors of bacteriophage clinical trials include health systems from around the globe; industry sponsors in the United States and France are currently responsible for most ongoing trials. Bacteria of interest vary based on which disease is being studied. However, most trials aptly focus on either *E. coli* or the ESKAPE pathogens (primarily *S. aureus*, *K. pneumoniae*, and *P. aeruginosa*) which represent over 60% of the target species (Figure 3). Target diseases mirror those patient populations in which recurrent or multi-resistant bacterial infections are expected, including urinary tract infections (UTI), prosthetic joint infections (PJI), bacteremia, and respiratory infections related to cystic fibrosis (Figure 4).

#### 5.2.1. Skin and Soft Tissue Infections (SSTI)

Eight clinical trials listed on clinicaltrials.gov evaluate SSTIs. All of these are phase I or phase I/II. Trials evaluate various phage applications, including four related to ulcers (i.e., diabetic foot ulcers, venous leg ulcers, pressure ulcers), two related to wound infections, one related to atopic dermatitis, and another related to the safety of the application of phage products directly to the skin. Of the eight SSTI trials, four studies have been completed. The other four are listed as ‘not yet recruiting’ [92,93,94,95]. The Phagoburn trial was a phase I/II trial evaluating a cocktail of phages versus standard of care to treat *P. aeruginosa*-infected burn wounds. The trial was terminated early due to insufficient efficacy in the treatment arm. It is believed that the failure of this study was due to the low titers (10^2^ PFU/mL) of the phage during treatment caused by either manufacturing or formulation issues [96,97]. The REVERSE study (phase I/II) looked at a five-phage cocktail targeting *S. aureus*, *P. aeruginosa*, and *A. baumannii* (TP-102), which was applied topically at 10^9^ PFU/mL/cm^3^ [98]. The treatment arms were TP-102 and standard of care versus placebo and standard of care. In a published preliminary assessment regarding the safety of the preparation, no severe adverse events were observed; the final efficacy data are pending [99]. The third SSTI trial (WPP-201) with available results is a phase I study published in 2009 evaluating the safety of phage therapy for the treatment of venous leg ulcers, targeting *S. aureus*, *P. aeruginosa*, and *E. coli* [100]. Participants were treated for 12 weeks with either saline or phage cocktail. The study concluded that the product is safe. No phase II study on the efficacy of the treatment has been identified. The fourth study on SSTIs utilized an AB-SA01 phage cocktail applied topically with increasing titrations [101]. The study’s results could not be found, but other studies which used the same or similar product from the company will be discussed later. The remaining SSTI studies listed online are not yet recruiting or ongoing.

#### 5.2.2. Lung Infections

A total of seven current clinical trials involve lung infections, with the majority (4/5) concerning infections related to cystic fibrosis (CF) [102,103,104,105,106]. All five CF studies specifically target *P. aeruginosa*. There is an extensive involvement of the Cystic Fibrosis Foundation in supporting these clinical trials, as *P. aeruginosa* is a significant cause of morbidity and mortality in CF patients [107]. Three of these CF trials are currently recruiting participants [102,103,104]. The MUCOPHAGES trial looked at the effect of 10 pseudomonas-targeting bacteriophages on induced sputum in 59 CF patients [108]. No results were published for this study which concluded in April 2012. Two of the lung-targeted phage trials named Tailwind and SWARM-Pa are for the same product, an inhaled *P*. *aeruginosa* phage cocktail named AP-PA02 [105,106]. The Tailwind study is a phase II clinical trial currently recruiting and looking at the efficacy, safety, and kinetics of AP-PA02 in non-CF bronchiectasis and chronic *P. aeruginosa* infections [105]. The SWARM-Pa is a phase Ib/IIa safety and tolerability study of inhaled AP-PA02 in CF patients with chronic *P. aeruginosa* infections which concluded in March 2023 and has not yet published its findings [106]. Another study currently investigating an inhaled *P. aeruginosa* bacteriophage product BX004 has concluded part one of their phase I/II study with results expected to be published in 2023. The NIAID is sponsoring a multicenter phase Ib/II study for a *P. aeruginosa*-specific IV-administered phage cocktail, WRAIR-PAM-CF1, determining safety and log reduction as pseudomonas sputum counts. This study is scheduled to be completed in 2024 [104].

An expanded access IND was granted for COVID-19-positive patients with secondary pneumonias caused by *A. baumannii*, *P. aeruginosa,* or *S. aureus*. The expanded access is no longer available as of December 2021, and no results have been published [109]. Lastly, an ongoing study explores the effect of sextaphage versus octenicept versus saline at preventing ventilator-associated pneumonia [110]. The study is ongoing and set to conclude in late 2023.

#### 5.2.3. Gastrointestinal (GI) Infections

According to clinicaltrials.gov, five clinical trials target the GI tract, two of which have completed. One trial evaluated the effect of a phage cocktail targeting *E. coli* in the gut on inflammation and gut microbiome in healthy adults [111,112]. Results indicated that the phage did not affect overall microbiome diversity. There was an average reduction in *E. coli* reads by 40% with no change in stool fatty acid production, lipid metabolism, and inflammatory markers. A reduced circulating IL4 was noted, speculated to be associated with an allergic or autoimmune response. The second GI phage trial study results have thus far only been presented as a poster at the AASLD meeting in 2021. Results presented the safety, tolerability, and pharmacokinetics of a *K. pneumoniae* administered orally and twice daily for three days in fourteen participants versus four placeboes. Results revealed an increase from 0 to 103 PFU, which was sustained through day 6 when they stopped collecting stool samples and concluded that the product is tolerable and safe. Notably, there was a 42.9% and 50% treatment-emergent adverse effect (TEAE) for treatment and placebo, respectively, and no treatment-related adverse effects were found [113,114].

The three other GI clinical trials are currently evaluating the safety and efficacy of a phage cocktail (ShigActive) in treating shigellosis, fecal bacteriophage transfer for GI maturation in preterm infants, and the safety and efficacy of an adherent invasive *E. coli*-specific phage cocktail in patients with inactive Crohn’s Disease [115,116,117].

#### 5.2.4. Genitourinary (GU)/Urinary Tract Infection (UTI)

Six clinical trials evaluate phage preparations in treating infections within the genitourinary tract for asymptomatic UTIs. Five are specific to UTIs, and one is looking at bacterial vaginosis. Two of these trials have been completed [118].

The first trial looked at the efficacy of Pyo phage—a treatment available commercially through the Eliava Institute in Tbilisi, Georgia, targeting S. aureus, *S. pyogenes*, *S. sanguis*, *S. salivarius*, *S. agalactiae*, *E. coli*, *P. Aeruginosa*, *P. Mirabilis,* and *P. vulgaris*—in treating urinary tract infections in patients undergoing the transurethral resection of the prostate. The study participants were randomized in a 1:1:1 ratio to intravesicular Pyo phage, standard of care, or placebo bladder irrigation, respectively. The results concluded that the phage was non-inferior to either standard of care or placebo [118,119]. The authors concluded the low treatment titers may have led to the lack of efficacy seen and that phage concentration at the site of infection as well as pathogen load are each important factors when designing phage therapy regimens.

The second completed UTI study was phase Ib and utilized a CRISPR-Cas3-enhanced phage cocktail targeting *E. coli* (LBP-EC01) [120]. The trial studied phage pharmacokinetics, pharmacodynamics, and safety profile when administered via the intraurethral route in 36 patients divided into a 2:1 ratio of treatment arm to placebo arm [121]. This same treatment is involved in one of the other listed UTI clinical trials, which is currently recruiting for phase II/III clinical trials concerning the efficacy of different LBP-EC01 doses for acute uncomplicated UTI caused by multi-drug-resistant *E. coli* in female participants [122].

Two other studies are phase I and phase I/II studies examining UTIs. One is active, not recruiting, and is looking at the safety and efficacy of their phage product on *E. coli* and *K. pneumoniae* UTIs [123]. The other trial is not yet recruiting and is a single-patient trial investigating a three-phage cocktail’s capacity to prevent recurrent drug-resistant UTIs [124]. The last is a phase III clinical trial examining the effect of a combination therapy of an *E. coli*-specific phage and prebiotic cocktail on bacterial vaginosis [125].

#### 5.2.5. Prosthetic Joint Infection (PJI)

Three PJI clinical trials are ongoing, per clinicaltrials.gov. Two of these studies share a sponsor and utilize a bank of bacteriophages with varying coverage against *S. aureus*, *S. epidermidis*, *S. lugdunensis*, *Streptococcus* spp., *E. faecium*, *E. faecalis*, *E. coli*, *P. aeruginosa*, and *K. pneumoniae*. Both trials utilize the experimental treatment in combination with a standard of care that consists of debridement, antibiotics, and implant retention (DAIR). ACTIVE 1 is a phase I/II open-label study investigating the feasibility of intraoperative versus intravenous administration and efficacy in first-time, chronic prosthetic joint infections of the knee or hip [126]. ACTIVE 2 is a phase II/III blinded study comparing debridement, antibiotics, and implant retention (DAIR) with and without bacteriophage therapeutics in participants with knee/hip PJI and prior surgery [127]. The third PJI clinical trial is a blinded phase II study evaluating the efficacy of phage(s) targeting *S. aureus* versus placebo administered intraoperatively in knee and hip DAIR procedures [128].

#### 5.2.6. Bacteremia

There are two ongoing bacteremia phage clinical trials. One of these is a phase I study evaluating the safety, tolerability, and pharmacodynamics of three different doses of an *E. coli* phage that was genetically modified to deliver a CRISPR/Cas payload (SNIPR001) administered twice a day for seven days [128]. DiSArm is a phase II clinical trial evaluating the safety, tolerability, and efficacy of a *S. aureus*-specific phage cocktail (AP-SA02) as an adjunct to antibiotics for subjects with *S. aureus* bacteremia [129].

#### 5.2.7. Other Infections

There is an open-label phase III study exploring the efficacy of nebulized Pyo phage, targeting *Staphylococcus*, *Enterococcus*, *Streptococcus*, *enteropathogenic E. coli*, *P. vulgaris*, *P. mirabilis*, *P. aeruginosa*, *K. pneumoniae*, and *K. oxytoca* in treating acute tonsillitis in children and adolescents [130]. There is an ongoing phase II study with regard to the safety and efficacy of a personalized *S. aureus* phage treatment for diabetic foot osteomyelitis versus a placebo at a 2:1 ratio, respectively [131]. Furthermore, a current preventative study compares the efficacy of oropharyngeal decontamination using a phage solution (targeting *Staphylococcus*, *Streptococcus*, *P. vulgaris*, *P. mirabilis*, *P. aeruginosa*, *enteropathogenic E. coli*, and *K. pneumoniae*) with an antiseptic (octinicept) and placebo (saline) in preventing respiratory-associated pneumonia [110]. The final open-label study listed terminated in 2010 in Poland: it was an open-label study examining whether bacteriophages could be used to treat patients with non-healing postoperative wounds, osteomyelitis, upper respiratory tract, genital or urinary tract in whom antibiotics have failed. The researchers concluded that phages were well-tolerated and have the potential to treat patients with otherwise untreatable chronic bacterial infections [77].

In addition to the interventional trials described above, there are several observational cases and cohort studies, bacteriophage banking programs, and microbiome research projects being conducted to better understand the role of phages and antibiotics in a healthcare setting. In summary, the rapid increase in the number of clinical trials in recent years is encouraging. The progression of more studies beyond phase I indicates an encouraging potential for phage-based treatments. The rigorous statistics, controls, dosing strategies, and treatment methods that were previously missing are now being developed to identify which patients will benefit most from these novel therapeutics. Much progress has been made in recent years because of the knowledge gained throughout the SPINDs, case studies, and lessons learned from published negative results.

In addition to an increase in clinical trials, many institutes have recently used compassionate use emergency INDs, expanded use, and temporary use authorizations to administer bacteriophage therapy to patients with no other therapeutic options. In a majority of cases when compassionate use of bacteriophage therapy is administered, an improvement in clinical symptoms is observed across multiple infection types [73,132]. Unfortunately, many compassionate use cases fail to thoroughly outline their methodologies or analysis, which may vary widely from patient to patient or between institutes, challenging an assessment of the efficacy of phage therapy overall. As additional clinical trials are initiated that employ standardized treatment and data collection protocols, we may likely see valuable efficacy data on the use of phages; for those phage therapy treatments that are shown to be effective, we can begin industrial-level production to utilize phage therapeutics more broadly.

### 5.3. Ethical and Regulatory Considerations

The use of bacteriophages as biological products with a therapeutic function fall within the scope of pharmaceutical legislation, and their development will need to comply with strict regulatory requirements, including the generation of robust and comparable scientific data [132]. Such criteria are important for the research and development (R&D) of medicinal products; however, these regulations can make performing such trials more costly and challenging to execute. Furthermore, several aspects of phage biology, such as their ability to transfer genes between bacteria and lysogenic conversion, still need careful elucidation to standardize the future industrial-scale production of phage pharmaceuticals [133].

Phage products are currently not commercialized, creating little incentive for authorities to develop bacteriophage-specific regulatory measures. Furthermore, FDA and EMA regulations stipulate that once a medicinal product is registered and approved, further modifications and improvements can only be applied to the final product following a new approval process. The production process for bacteriophages may prove to be challenging to standardize, primarily due to phages’ propensity to evolve as well as the co-evolutionary dynamics between phages and their bacterial hosts (which can result in mutations and changes in virulence). Even after the removal of endotoxin and the careful production of phage cocktail preparations, the longer that phages are left in solution, the more likely they are to interact with each other, become contaminated, or otherwise undergo changes that may cumulate in a compositional change of the resultant product. In addition, phage–bacterial interactions are complex and can cause resistance mechanisms such as phenotypic shifts and point mutations in surface structures that are recognized by phages as receptors [134].

Compassionate use applications of phage therapy are being studied under the stipulations set out by the Declaration of Helsinki, Article 37 (Unproven Interventions in Clinical Practice). The process and conditions for using this form of phage therapy are controlled by each nation’s specific regulatory agencies, such as the Food and Drug Administration (FDA) in the United States, the Therapeutic Goods Administration (TGA) in Australia, the European Medicines Agency (EMA) in the European Union (EU), or the Brazilian National Health Surveillance Agency (Anvisa) in Brazil [132]. In contrast to larger clinical trials, the compassionate use of specific therapies is primarily aimed at monitoring, diagnosing, or treating patients and not at obtaining efficacy and safety data to support the licensure of any investigational product.

However, there are requirements for compassionate use, these being: (i) the patient must have a serious or immediately life-threatening condition or disease; (ii) no comparable or satisfactory alternative therapeutic options may exist; (iii) the potential benefit of the therapy must justify the potential risks of its administration; and (iv) the provision of the product through compassionate use must not interfere with or compromise other forms of clinical investigations that may be necessary to support the licensing of the product.

The search for unapproved therapies through compassionate use has gradually gained momentum over time. Families have requested access to these therapies and many have received legal and financial support for phage treatment through social groups [135]. Considering this, some countries have legislation in place to support this type of care for the critically ill, such as the “Right-to-try” law in the United States [136]. In Poland, in contrast, phage therapy is considered an “Experimental Treatment”, covered by the adapted law of 5 December 1996, for use by medical professionals [137]. The HIIET has pioneered the compassionate use of therapeutic phage in modern medicine, significantly advancing research on therapeutic phages [138]. In a similar effort to address the lack of specific regulations concerning the preparation and use of phages for therapeutic purposes, the Belgian Federal Agency for Medicines and Health Products modified its own permits to allow the use of phages as an active ingredient in pharmaceutical compounding (also known as magistral preparations). This allows phages to be prescribed individually and produced following internal guidelines [139,140]. Moreover, phage therapy does not follow a “one size fits all” standard but is tailored according to patients’ needs, making it difficult to establish uniform regulation [141].

The case reports on phage applications in compassionate use circumstances have only addressed a small proportion of antibiotic-resistant cases [142]. Notably, most such cases are geographically concentrated around already-established testing centers or are conducted by a few physicians and researchers with the necessary expertise to strategize and carry out the treatments. Expansion of this clinical practice will not likely occur until marketing approval is attained. While clinical trials are ongoing, research centers must work in tandem to optimize phage availability, transportation logistics, phage purification, data summarization and publication, and the production of clinically stable and applicable phage formulations [143].

Although phage therapy presents potential advantages over currently available therapies and has a promising market, an analysis of the current framework reveals a lack of targeted regulation by major national and international agencies [134]. This lack of established regulatory standards is largely due to limited availability of structured, in vivo evidence of successful phage therapy; the limited body of evidence is itself due to the lack of established clinical trials that follow national and international ethical standards [144,145].

These ethical standards are nevertheless critical, as phage research appears to be following the larger trend of globalization that increasingly places clinical studies in LMICs. As novel therapeutic development research in the United States faces more substantial scrutiny from regulatory agencies such as the FDA, there is a growing tendency to export clinical trials to other countries with less stringent policies for human research and where operating costs for clinical research are generally lower. This practice raises concerns regarding the potential exploitation of local residents in LMICs who may enroll in such studies for a perceived financial or health benefit, facing the risks of clinical trials while the payoff of the research ultimately benefits people living in HICs. This concern has been substantiated by the historical precedent set by ‘overseas’ clinical experimentation with certain contraceptives (e.g., Norplant studies in Bangladesh) and HIV medications, among various other pharmaceuticals [146,147,148,149]. Although clinical trials in LMICs can provide healthcare to marginalized communities that may otherwise have limited or no access to specific health services, more data will be critical to further evaluate and confirm the acute and long-term safety profiles for phage therapeutics.

Despite these important concerns, it can be argued that insofar as LMICs tend to contain a disproportionate number of bacterial species with AMR capacities, there is particular value in conducting phage research in regions where clinical research findings may be most generalizable to the larger population and where phage therapy stands to confer the greatest benefit [150]. Moreover, the phages that target these AMR bacteria are posited to be more likely found in the same regions where these bacteria emerged, guiding investment in conducting phage isolation and characterization where there is a greater diversity of phages to be discovered. Conducting research in LMICs enables a more sustained investment in R&D infrastructure for relevant diseases and therapeutics, ideally facilitating a more affordable and efficient process of providing these drugs to the communities most in need. In contrast to many other pharmaceuticals that are manufactured in HICs and sold to LMICs at marked-up, cost-prohibitive prices, phage therapeutics have the potential to be more affordable and accessible to marginalized communities when the research and drug development are conducted locally. Nevertheless, the concerns of conducting research involving socioeconomically disadvantaged populations warrant that the utmost precaution must be applied when recruiting these study participants and that all participants undergo a culturally sensitive and thorough process of acquiring informed consent prior to enrollment. These same principles apply to clinical research performed on marginalized communities within HICs.

## 6. Potential and Limitations of Phage Therapy

### 6.1. Benefits and Potential

The safety profile and tolerability of phage therapy have thus far been encouraging. Inherently, bacteriophages do not pose a risk to the health of eukaryotic cells which lack the necessary surface receptors to be infected by these viruses [1]. Potential adverse reactions to bacteriophage administration are typically due to a by-product of their isolation in large concentrations, such as endotoxins released from contaminant Gram-negative bacteria, rather than the phages’ properties. Various protocols exist to remove endotoxins from phage preparations successfully [151]. Previous reviews have indicated that severe toxicity—manifesting as hypersensitivity reactions or cytokine storms characterized by symptoms such as fever, wheezing, and shortness of breath—resulting from phage administration is rare [73,152]. Although it is difficult to ascertain causality, there have been rare cases in which adverse effects have occurred during phage administration, including transaminitis, hypotension, nausea, fevers, and chills [153,154]. In contrast, many broad-spectrum antibiotics used in clinical practice today carry substantial side-effect profiles and can lead to life-threatening organ damage or toxicities that complicate their use [155,156].

There are limited data regarding public understanding of bacteriophages and patient perceptions of phage therapy. This is consistent with phage therapeutics being a relatively novel frontier in clinical infection management. The first investigation into public knowledge and receptivity of bacteriophages was conducted in Scotland in 2020. Despite a relatively small sample size and limited statistical power, the study evaluated patient understanding of AMR and phage therapy following treatment for diabetic foot ulcers and provided valuable insight into patient receptivity to phage therapies. Most patients were aware of the concept of antibiotic resistance and expressed concerns about its worsening implications, but many were not aware of bacteriophages (heard of by less than 25% of participants) let alone phage therapy (heard of by less than 10% of participants). Patients were generally enthusiastic about phage therapy when framed as a potential tool to address AMR and were no more concerned with clinical phage applications than antibiotic therapy. Notably, the study also revealed common themes among patient questions and concerns [157]. For phages to be successfully re-introduced and advanced in a clinical setting, more studies are needed to explore patient perspectives and concerns with regard to this novel therapy. Patient-centered concerns and experiences will inevitably shape how these therapeutics are developed and utilized. Accordingly, additional studies should also be conducted to evaluate the thoughts and concerns of participants in phage-related clinical trials.

As these viruses are ubiquitous and can be easily isolated from environmental samples, an impressive reservoir of potentially efficacious bacteriophages has yet to be discovered. We have already seen encouraging results from phage therapy administration, achieving this with a limited bacteriophage arsenal relative to that which remains unidentified. Moreover, as there does not appear to be a single bacterium that “cannot be lysed by at least one bacteriophage”, the bacterial coverage of phage application stands to extend beyond what has been achieved with antibiotics [52]. Further discovery of new phages extends beyond those that are naturally occurring; a new frontier of phage research will stem from engineered and synthetic phages, created through homologous recombination-based and CRISPR-Cas techniques [158]. There is significant potential for genetic engineering of bacteriophages, with applications that include transforming lysogenic phages into purely lytic phages via prophage induction, extending host ranges, increasing antibiotic sensitivity, and improving lytic activity of phages via removal of repressor genes [159,160].

Methods of administering bacteriophage preparations to individual patients have varied based on the type of infection and have included intravenous, inhaled, injectable, and oral formulations. It is reassuring that favorable results have been seen with a variety of routes of administration [78]. Another benefit of bacteriophage therapy is the general ease of administration schedules compared to typical antibiotic regimens. Although past administration schedules varied more substantially, data from recent case reports support the administration of phage therapy every 12 h without requiring more frequent administration [161].

Bacteriophages supersede antibiotics in their ability to penetrate through biofilms and therefore have promising applications when used alone or as a combination therapy with antibiotics to treat PJIs. Multiple case reports detail the clinical improvement of PJIs following phage therapy administration [162,163,164]. In vitro studies also demonstrate the promising effects of bacteriophages against biofilms and the potential impact that bacteriophage/antibiotic combination therapy may play in decreasing the likelihood of evolving phage resistance [165]. Biofilms are ubiquitous in PJIs because the “foreign” material (e.g., titanium rods, screws, plates, etc.) introduced to the body during these surgeries serves as a nidus for biofilm production. PJIs are projected to cost more than USD 57.0 million annually by 2030 [166]. While typical antibiotics are largely unsuccessful due to their difficulty in penetrating biofilms, bacteriophages subvert these biofilms via polysaccharide depolymerases and thereby increase sensitivity to the antibiotics [167].

Although the specificity of bacteriophages may pose challenges to industrial-scale manufacturing and distribution, it is precisely this specificity which makes them so invaluable. With the administration of antibiotics, pathogenic bacteria are eradicated at the expense of non-pathogenic, commensal bacteria and subsequent disruption of the microbiome. The most infamous consequence of this broad bacterial annihilation is a proliferation of *Clostridium difficile* leading to pseudomembranous colitis. Among hospital-acquired infections, *C. difficile* is currently one of the most significant and leads to extended hospital stays, increased expenditures, and compromised patient well-being with estimates of annual costs in the U.S. of USD 6.3 billion [168,169]. As bacteriophages have narrow specificities and rarely target more than one bacterial type, they allow for the eradication of pathogenic bacteria while sparing commensal organisms and maintaining a healthy microbiome [170]. However, phages and antibiotics are currently co-administered when applied in the clinical setting, and phage-antibiotic in vitro synergy studies have shown promising results [89]. Compared with isolated bacteriophage administration, phage therapeutics combined with antibiotics have demonstrated better bactericidal activity and biofilm eradication against several bacterial species, including *P. aeruginosa*, *E. coli*, *and S. aureus* [88,171,172,173].

The co-administration of antibiotics and phages typically results in an additive or synergistic effect in which their interaction confers an effect that is equal or greater, respectively, to the sum of their individual effects. However, combination therapy has also been shown to produce a permissive drug interaction in which a phage that has not previously demonstrated any bactericidal activity becomes activated upon co-administration with an antibiotic [174]. The synergistic relationship between specific phages and antibiotics has been best exemplified by studies that identified an upregulated phage production by the bacterial host cell following treatment with sub-lethal concentrations of antibiotics that accelerate the lysis of infected bacterial cells and therefore facilitate phage propagation [175]. This effect has been observed with a myriad of antibiotic and phage combinations, including T4-like phages and beta-lactam antibiotics, quinolone antibiotics, and mitomycin C [175].

Numerous studies have illustrated the potential cost-effectiveness of bacteriophage therapy, especially in comparison with antibiotics [176]. As discussed previously, the relatively low cost and ease of phage isolation makes them an ideal alternative antimicrobial for treating bacterial infections in LMICs. The benefit of incentivizing isolation and characterization of bacteriophages in lower-resource regions is two-fold. Investing in the ability of LMICs to independently produce their own bacteriophage bank may decrease the necessity to purchase and import marked-up phages from other countries. Furthermore, discovering bacteriophages in the regions where they will most commonly be used in a clinical setting has tremendous potential merit. Due to selective pressures in a shared geographic region, both a phage and a bacterial species in that area may co-evolve with one another. This results in phage-bacteria sensitivity and interplay, which may not otherwise be seen in geographically distinct environments [177]. As international collaboration in this field increases, it will be invaluable to compare the efficacy of bacteriophages from one nation against clinical bacterial isolates from another and vice versa. Different academic labs with their own specific focus on bacteriophage research and phage bank preparation are increasing around the world and are expected to continue increasing over the next several years. An overview of the main university laboratories and research institutions working on phage therapy worldwide can be found in the Appendix A.

### 6.2. Limitations and Future Directions

Bacteriophages present an attractive and necessary potential therapy, especially in light of growing antimicrobial resistance on an unprecedented global scale. International health agencies must aim to optimize the timely, comprehensive, and appropriately regulated provision of antibiotics to LMICs. Numerous aforementioned variables contribute to limited healthcare infrastructure in LMICs, underlying the inequitable distribution and erroneous utilization of antibiotics which increase the selective pressures for bacteria that develop AMR [11,178,179]. Before solely prioritizing phage therapeutics, the international community must take a preventative approach to combatting AMR by addressing root causes of inappropriate antibiotic use. HICs, including the U.S., Canada, and United Kingdom, have detailed their own domestic strategies for addressing AMR, but largely stop short at extending those resources to LMICs. Organizations such as the Global Antibiotic Resistance Partnership (GARP), which aim to develop actionable policies that mitigate AMR in LMICs, have been successfully established in a few scattered countries in Africa and Asia; however, further international collaboration is needed to comprehensively address this global health crisis.

Previously, we reviewed the legal and regulatory hurdles involved in conducting bacteriophage clinical trials and in applying phage therapeutics in clinical settings. However, the adoption of phages as a well-supported and routine therapy is further challenged by biological limitations. The efficacy of bacteriophages in vivo has been shown to depend on an adequate immune response. Appropriate immune activation, primarily a robust neutrophil response, may confer a synergistic effect that tempers phage resistance and is critical to the resolution of an acute infection [180]. In immunocompetent hosts, phage-resistant populations of bacteria are cleared by host innate immune processes; in contrast, neutropenic patients lacking sufficient clearance mechanisms have been shown to be at a higher risk of treatment failure due to a lack of the synergistic neutrophilic elimination of phage-resistant bacteria. Additional analyses indicated that even among infections in which the bacterial population was entirely phage-sensitive, host immune clearance mechanisms would still likely be required to completely eradicate the offending pathogen(s) [180]. The implications of host-immune-system–phage interactions will require further elucidation.

Phages are self-multiplying, which creates an additional difficulty in determining the best dosage, as actual phage concentration at the infection site may differ from the density of phage administered in a controlled manner, especially when co-administered with certain antibiotics that can potentiate phage propagation via enhanced cell lysis of phage-infected bacterium [181]. Further pharmacokinetic and pharmacodynamics studies are needed to better understand the implications and practicality of dose standardization.

Another potential concern of using bacteriophages in the clinical setting is the development of bacteriophage resistance among the targeted bacteria. Evidence of decreased phage efficacy due to acquired viral resistance has been demonstrated among numerous bacterial species in animal trials [182]. Resistance to phages is produced through various bacterial mechanisms, including adsorption resistance (e.g., phage receptor modulation or increased extracellular matrix synthesis), restriction-modification, and abortive infection, among others [182,183,184]. Clinically, disease improvement at the beginning of phage therapy, followed by deterioration and infection relapse, may partially contribute to this resistance acquisition [73]. However, rarely are bacteria found to cultivate phage resistance while remaining impervious to host immunity—evolutionary pressures that select for phage resistance appear to leave some bacterial species increasingly vulnerable to host clearance and immune defense mechanisms, as well as to antibiotics [182]. Therefore, phage resistance acquisition presents a sort of bidirectional action in that this limitation also strengthens the potential of other therapies. When bacteria develop resistance to phages by further mutating receptors that were previously mutated to confer their antibiotic resistance, these bacteria inadvertently become more antibiotic sensitive [185,186,187]. Further research into the potential relationships between phages and different antibiotic classes is required to better understand these events and determine which antibiotic/phage combinations may be best optimized for clinical application [86].

To combat the development of bacterial resistance to bacteriophages, phages are now commonly administered as “cocktails”: preparations containing numerous phages with different mechanisms of action. Multiple studies have revealed a decrease in the acquisition of phage resistance in bacteria treated with more than one phage at a time [184,188,189]. This preference for administering multiple phages at once means that an extensive bacteriophage bank is necessary for generating phage cocktails with effective clinical utility and requires copious sensitivity testing of the specific clinical strain of bacteria against various phages known to infect that bacterial genus. Therefore, routine isolation and genetic engineering of novel bacteriophages via homologous recombination will prove necessary to mitigate resistance against phages in the clinical setting as they become applied more broadly and with greater frequency. These processes can be both time-consuming and costly [190]. National bacteriophage banks with large reservoirs of potential phages are predicted to alleviate this challenge and will be further augmented by international collaboration amongst national banks [177].

Collaboration amongst phage research institutions has proven to be a cornerstone of successful bacteriophage therapy and research development [191]. The first case of phage therapy in the U.S. which led to the development of the Center for Innovative Phage Applications and Therapeutics (IPATH), occurred at a time when the University of California San Diego Medical Center did not have a phage bank or associated phage laboratory but was rather sustained by multi-institutional cooperation, namely with Texas A&M University. This support was garnered primarily by personal outreach explaining the clinical case in need of phages [192]. This trend was then mirrored internationally, with the first case of phage utilization in Israel resulting from a collaboration between Adaptive Phage Therapeutics and the U.S. Naval Medical Research Center, further supplemented by the then established IPATH laboratory [193]. Aslam et al. described a total of 785 requests for bacteriophage therapy from both U.S.-based and international physicians and patients within a timeline of two years [73]. While only a small proportion of these cases resulted in phage therapy administration, it is evident that multi-center collaboration is necessary for phage therapy because many institutions do not yet have their own phage banks or appropriate experience with phage administration in a clinical setting. These cases, among others, exemplify how successful collaboration can be achieved through open lines of communication and support both of and from individuals with well-established credibility within the scientific and medical communities. Despite these instances of successful multi-center collaboration yielding positive outcomes from cases of phage administration, it will be necessary to expand access to phage therapeutics and improve avenues for transparency amongst international institutions to promote inclusivity in the field, especially when considering LMICs, rural communities, and smaller institutions.

While efforts such as the Africa Phage Forum are promising examples of collaboration amongst nations within the same global region, more work is needed to increase collaboration between distinct global regions, specifically with regard to LMICs [194]. Currently, most publications related to bacteriophage research have originated from the U.S. and China, and the most productive countries conducting phage therapy research related to treating bacterial infections in humans include the U.S., China, Canada, India, Poland, Spain, Australia, and South Korea. Furthermore, international cooperation maps have indicated frequent collaboration amongst industrialized nations, with a notable lack of involvement of LMICs [191]. This is especially distressing considering the incredible opportunity phage therapy presents to improve global health equity, especially if the production of distinct national/regional phage banks is incentivized. Countries in South America, for instance, contain a large potential reservoir of novel phages; however, investment in phage therapy research has been limited. Despite this limited investment in phage research across the continent, phage research facilities in Brazil are working toward developing a Brazilian bacteriophage bank to serve as a reservoir for future phage therapeutics and as a data center for collaboration in international phage research. Brazil is the largest country in South America and contributes substantially to the economic landscape of the continent. By sharing geographical proximity and significant trading relationships with several neighboring LMICs, Brazil has an immense potential to serve as a critical phage research hub in South America.

In-person phage conferences and the exchange of personnel/equipment for the development of phage projects are undoubtedly valuable for advancing phage research worldwide. Teleconference frameworks such as Project ECHO allow for invaluable expertise to be shared amongst medical professionals, even extending into remote geographic areas. By following such a framework, information related to phage research may be easily shared amongst institutions regardless of their geographic location, and more importantly, may be guided by the specific needs of those participating. This knowledge sharing is a crucial aspect of international phage collaboration, as this therapy represents an opportunity for LMICs to sustainably develop their own national phage banks rather than rely on phages shipped from industrialized nations as is seen frequently with other therapeutics [177].

Nagel et al. discusses the importance of collaboration between neighboring countries to mount timely responses to regional outbreaks, the need for phage-related contracts and licensing agreements, and the consideration of phage products as contributing to the public good as opposed to being seen as profit-garnering products controlled by private enterprises [177]. For phages to be effectively and efficiently transported between countries and regions, there must be a well-established, regulated, and frequently updated resource detailing both the full scope of isolated phages stored in various banks around the world as well as their respective host ranges. Currently, national and institutional updates are published sporadically in a variety of scientific journals, and there is no single standardized outlet for combining all current phage information [195,196]. While databases such as the Phage Directory exist, contribution to the site is voluntary and independently motivated, which results in an under-approximation of the global scale of phage labs and phage-oriented scientists. While independent institutional or national updates are valuable, the future of phage therapy will depend on a more robust and accurate database which is managed in a structured and standardized manner, regardless of institutional association.

Numerous protocols currently exist for phage isolation and purification, but sustainable international collaboration will depend on standardization to ensure quality control of phages and phage products being shipped from one institution to another. While the benefits of international collaboration are promising, the formal construction and regulation of such cooperation will be inevitably challenging. The current method of collaboration is based heavily on pre-existing, independent institutional, personal, or professional contacts, and there are few guidelines and minimal formal structures in coordinating information sharing and laboratory training. This current method of collaboration confers some benefits in that it is free of the secondary hurdles which the introduction of regulatory agencies and committees frequently produce. However, a balance must be achieved amongst international phage research labs in deciding regulatory priorities and standards related to disseminating phage products as well as to the training of personnel and identifying intellectual property. The World Health Organization’s Global Observatory on Health Research and Development, which was launched in 2017, has highlighted the disparities in research funding between HICs and LMICs and concluded that there is a lack of collaboration between HICs and LMICs, while most collaborations occur between similar income groups [197]. This information highlights disparities that undoubtedly challenge the phage therapeutic production aspect of phage research in some areas of the world and emphasizes the need to encourage collaboration between high- and low-income nations, as well as the investment of international research funding agencies in infrastructure development.

It is not simply international collaboration which will be necessary to further and improve phage research; transdisciplinary cooperation also contributes to the successful development of this therapy [198]. There is much to learn from past and ongoing research involving phage application in agriculture, and a robust reservoir of in vivo animal studies involving phage application can be garnered from this field. For instance, the use of phage preparations in pre-slaughter practices to decrease pathogenic bacterial contamination of meat for human consumption has been shown to be effective for a variety of bacteria (including *Salmonella*, *Campylobacter*, *Listeria*, and *E. coli*) which can otherwise lead to serious disease in consumers [199]. These studies have revealed similar results related to single-phage versus phage cocktail administration in its relation to phage resistance acquisition. Routes of administration which have already been evaluated in animals, including livestock, have included oral, rectal, and intramuscular, among others [200,201,202]. Even non-food-producing animals, such as companion animals, are reservoirs for resistant bacteria and have been the subject of phage research that has produced valuable data regarding phage pharmacokinetics and efficacy [203,204]. In light of this One Health approach, the role of phage therapy in combatting antimicrobial resistance pivots from that of a strictly clinical perspective to include the application of phage in the agro-food and veterinary sector, where the majority of antibiotics are used worldwide [177,198].

Although the number of in vitro antibiotic/phage synergy studies are increasing, most compassionate use cases of phage therapy administration do not employ routine antibiotic and phage synergy testing [73,205]. The ARLG discussed various challenges and recommendations facing phage therapy via an expert panel in 2022. Future perspectives included the development of combined antibiotic and phage sensitivity testing methods which could be employed by clinical microbiology laboratories to assist phage-antibiotic selection in clinical cases [78]. The development of efficient and accurate protocols for such testing will be imperative for this to become standard practice. Furthermore, the results of synergy testing in the lab must be contextualized with their associated clinical outcomes [78]. Evaluating not only the synergy of phage-antibiotic combinations, but also the potential impact of administration route and combined preparations will be of interest as the production of phage preparations for clinical use increases.

The specificity of phages leads to difficulties when selecting and constructing phage cocktails; few phages have demonstrated efficacy against multiple bacterial families, which challenges treatment regimens for heterogeneous infections. More work is needed to improve the efficiency of this process via high-throughput methods of phage screening and genome engineering, as has already been initiated by some authors through the use of RNA-targeting CRISPR-Cas13a [206,207,208]. The potential of synthetic phages is excellent, however, streamlining the requisite techniques will enable their full benefit in adding to the phage therapy arsenal.

Furthermore, many bacteriophages are restricted in their efficacy against different strains of bacteria, even within the same species. Despite the input and relative success of phage cocktail administration, the potential variability in phage concentration for each unique phage in a cocktail indicates that phage resistance may remain a risk [209]. Additional methods in which resistance to phage therapy may be mitigated include switching the type of phage being used (i.e., selecting a phage known to utilize different methods of bacterial entry and multiplication) as well as selecting phages with more rapid bactericidal activity—often represented via a favorable adsorption rate and robust burst size and reproductive rate [209,210].

Additional practical limitations to the adoption of phage therapy include the labor and expenditures of phage discovery and research, the storage requirements unique to different phage preparations, and the challenges of administering specific bacteriophage cocktails. Phage isolation, while relatively straightforward in principle, varies between phages depending on the bacterial host of interest; some bacteria are inherently more challenging to manipulate in the laboratory setting [211]. In this sense, the discovery pipeline throughput will require substantial optimization if there is to be a successful scaling-up of this process for industrial manufacturing and administration. These difficulties in streamlining this process lead to increased delays in the bench-to-bedside transfer of necessary phages, with a median 170.5-day delay between request and administration to the patient noted by some authors [73]. However, there are already projects focused on optimizing the phage research process via software and technology platforms that facilitate the identification of prophages’ “on-demand” production of bacteriophages [212]. There is also significant variation in how phages are safely stored [213]. Each cocktail preparation has a unique shelf-life for stability that must be closely monitored to ensure that all phage preparations administered to patients are safe and effective [214]. Furthermore, acquired mutations among bacteriophages with extended storage times may pose another challenge to successful production. Although the starting product directly after manufacturing may be uniform, the gradual and sporadic development of mutations creates variations between cocktails otherwise comprised of the same phages, which can have significant clinical consequences upon administration to a patient [52]. Due to this variation and the extensive diversity of bacteriophages, regulating phage therapeutics will likely prove challenging, particularly because the therapies are most effective when tailored to a specific patient or disease state.

## 7. Conclusions

Rising antimicrobial resistance is a significant public health and global health concern. The imminent consequences of delaying novel therapeutic development are clear. Although antibiotics have served as the gold-standard therapy for treating bacterial infections for the past eighty years, the clinical relevance of this class of drugs is waning in the face of rapidly evolving MDR organisms. Novel antibiotic discovery needs to be more incentivized, as it currently yields a diminishing expansion of the available antimicrobial arsenal. As the number of patients succumbing to multi-resistant infections continues to increase yearly, bacteriophage therapeutics are gaining traction as a viable alternative antimicrobial therapy. Although the ability of these viruses to combat bacterial infections was acknowledged more than a century ago, the application of this naturally occurring antimicrobial has been hindered by various obstacles. Now, evidence of the efficacy of these bacteriophages is represented by numerous successful case reports among individual patients. However, clinical trials and R&D standardization are necessary to transition these investigational drugs to a common-place adjunct therapy to antibiotics. Adopting bacteriophage therapy into routine clinical practice will require committed and extensive investment by research laboratories and hospital systems, collaborating with international scientific and medical communities. This must include a consensus by regulatory agencies and a flexible perspective which will evolve alongside our understanding of the nuances of this potential therapy.

## Figures and Tables

**Figure 1 viruses-15-01020-f001:**
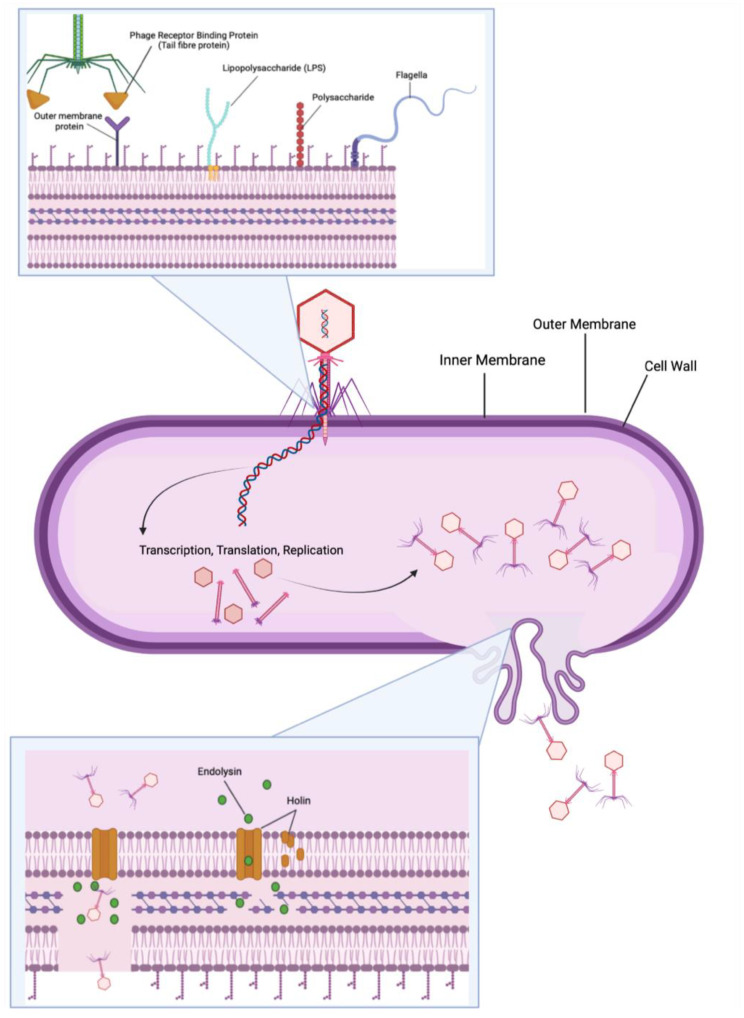
Bacteriophage adsorption, replication, and release of phage progeny. Attachment of bacteriophages to bacterial cell membranes occurs through a process known as adsorption, whereby phage receptor binding proteins (RBPs) located on the phage tail interact with corresponding, specific receptors on the bacterial cell surface. Receptors can include lipopolysaccharide (the most common receptor on Gram-negative bacteria), polysaccharides, proteins, flagella, and pili. Upon binding, genetic material is transferred from the phage head into the bacterial cell, and host molecular machinery is used to generate new phage progeny in the case of lytic bacteriophages. Phage release occurs via bacterial cell lysis, facilitated by holin–endolysin interaction, resulting in cell membrane breakdown and eventual cell death.

**Figure 2 viruses-15-01020-f002:**
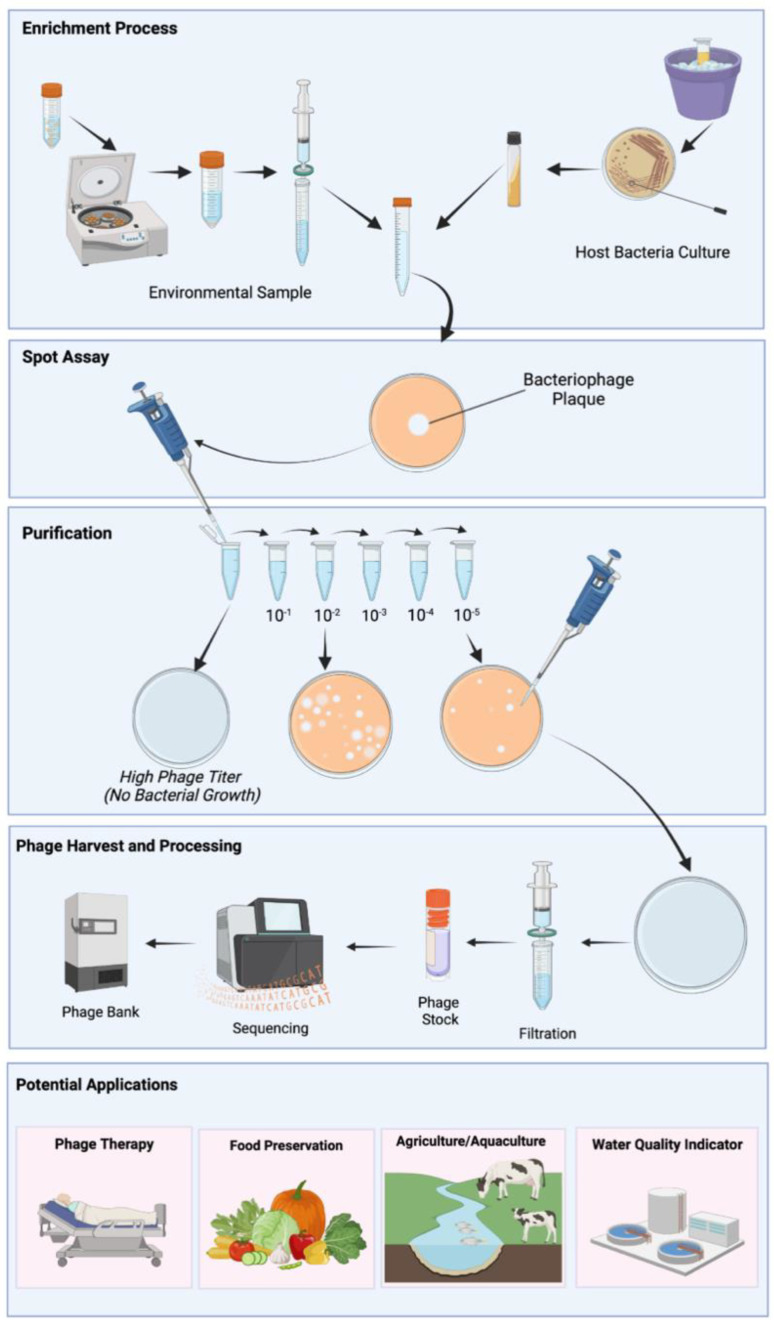
Process of bacteriophage isolation for therapeutic use. Phage isolation begins with environmental water samples that are filtered to remove bacteria. The resultant filtrate is combined with a bacterial culture of interest in a process known as enrichment, whereby the concentration of phages effective against the specific host bacteria increases. After enrichment, spot assay allows for visualization of phage presence on a solid bacterial lawn. Phages from the resultant phage plaque are combined with a neutral buffer, and serial dilutions are performed before further plating to obtain distinct phage plaques of homogenous morphology. After phage harvest, amplification, sequencing, and characterization are performed to determine phage novelty. Phages are added to bacteriophage banks and international phage directories to be later employed in different applications.

**Figure 3 viruses-15-01020-f003:**
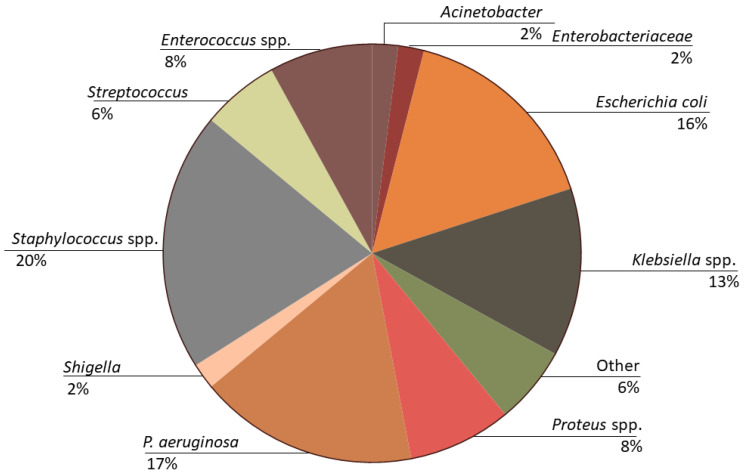
Bacterial species addressed by clinical trials. Many of the currently registered clinical trials on clinicaltrials.gov address an ESKAPE pathogen (*E. faecium*, *S. aureus*, *K. pneumoniae*, *A. baumannii*, *P. aeruginosa*, or *Enterobacter* species), predominantly *S. aureus*, *K. pneumoniae*, and *P. aeruginosa*. *A. baumannii*, *E. faecium*, and *Enterobacter* species are less frequently included. Other commonly targeted bacteria which are not considered ESKAPE pathogens include *Streptococcus* spp., *Proteus* spp., and *E. coli*. Bacterial targets denoted “Other” include *Burkholderia*, *Stenotrophomonas*, *Salmonella*, *Serratia*, *Citrobacter*, and *Morganella*. For some clinical trials, specific bacterial species were not defined on clinicaltrials.gov; therefore, some groupings include more than one species. Among the studied *Staphylococcus* spp., 51.7% are *S. aureus*, 9.5% *S. epidermidis*, or 9.5% *S. lugdunensis* as well as 23.8% undefined species of *Staphylococcus*. For *Enterococcus* spp., 25% are *E. faecium* versus 25% *E. faecalis* and approximately 50% undefined species of *Enterococcus*. For *Proteus* spp., 50% are *P. mirabilis* and 37.5% *P. vulgaris*, with 12.5% unknown species of *Proteus*. For *Klebsiella* spp., 76.9% are *K. pneumoniae* and 15.4% *K. oxytoca*, with the remaining 7.7% being undefined species of *Klebsiella*.

**Figure 4 viruses-15-01020-f004:**
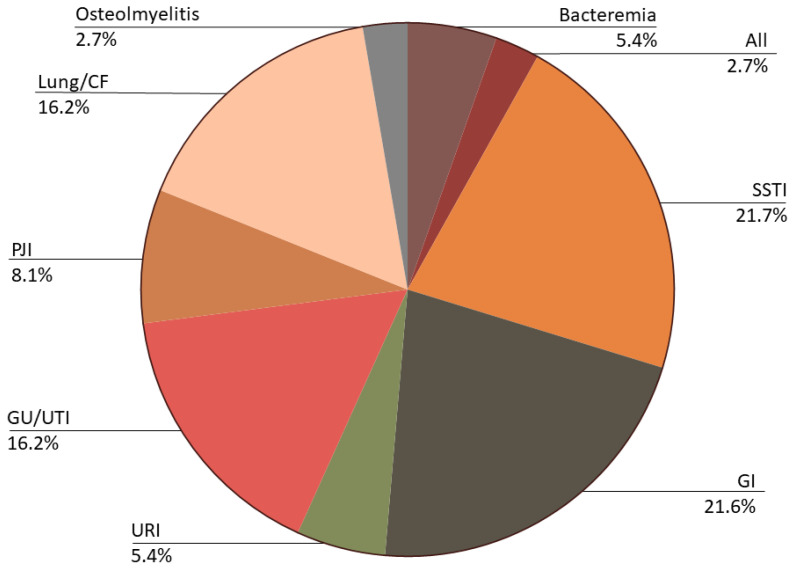
Indications for interventional clinical trials by infection site. A large proportion of currently registered bacteriophage clinical trials on clinicaltrials.gov (43.3%) involve the application of phage for either skin and soft tissue infections (SSTI) or gastrointestinal infections (GI). However, other infection sites, such as genitourinary tract infections (GU/UTI) and lung infections (including those in patients with cystic fibrosis) account for an additional 32.4% of the infections evaluated in these trials. The remaining trials involve patients with bacteremia, osteomyelitis, upper respiratory tract infections (URI), and prosthetic joint infections (PJI). A total of 2.7% of current phage clinical trials evaluate their use in treating non-healing wounds or infections of bones, upper respiratory tract, and genitourinary tract for which extensive antibiotic regimens failed or the use of a targeted drug was contraindicated.

**Table 1 viruses-15-01020-t001:** Current active or recruiting clinical trials involving phage therapy.

Study Title/Identifier	Condition or Disease	Microorganisms	Phase	Status	Sponsor/Collaborator
Cystic Fibrosis bacterioPHage Study at Yale (CYPHY)	Cystic Fibrosis	*P. aeruginosa*	1/2	Active, not recruiting	Yale New Haven Hospital New Haven, CT, USA
Bacteriophage Therapy in Tonsillitis	Acute Tonsillitis	*Staphylococcus* spp. *Enterococcus* spp. *Streptococcus* spp. *Enteropathogenic E. coli*	3	Active, not recruiting	Tashkent Pediatric Medical Institute Tashkent, Uzbekistan
Bacteriophage Therapy in Patients with Urinary Tract Infections	Urinary Tract Infection Bacterial	*E. coli* *K. pneumoniae*	1/2	Active, not recruiting	Adaptive Phage Therapeutics, Inc. (Washington, DC, USA)
Bacteriophages To Treat Liver Disease Eliminating Harmful Bacteria (BATTLE)	Alcoholic Hepatitis	*E. faecalis*	-	Recruiting	Copenhagen University Hospital Hvidovre Hvidovre, Denmark
A Phase 1b/2 Trial of the Safety and Microbiological Activity of Bacteriophage Therapy in Cystic Fibrosis Subjects Colonized with Pseudomonas Aeruginosa	Bacterial Disease Carrier; Cystic Fibrosis	*P. aeruginosa*	1/2	Recruiting	National Institute of Allergy and Infectious Diseases (NIAID)
Nebulized Bacteriophage Therapy in Cystic Fibrosis Patients with Chronic Pseudomonas Aeruginosa Pulmonary Infection	Chronic Pseudomonas Aeruginosa Infection; Cystic Fibrosis	*P. aeruginosa*	1/2	Recruiting	BiomX, Inc. (Ness Ziona, Israel)
Bacteriophage Therapy in Patients with Diabetic Foot Osteomyelitis	Osteomyelitis; Diabetic Foot Osteomyelitis	*S. aureus*	2	Recruiting	Adaptive Phage Therapeutics, Inc. (Washington, DC, USA)
Phage Safety Cohort Study (PHA-SA-CO)	Prosthetic Joint Infection; Severe Infection	*	*	Recruiting	Hospices Civils de Lyon (Lyon, France)
Phage Safety Retrospective Cohort Study (PHASACO-retro)	Bone and Joint Infection; Prosthetic Joint Infection	*	*	Recruiting	Hospices Civils de Lyon (Lyon, France)
Study to Evaluate the Safety, Phage Kinetics, and Efficacy of Inhaled AP-PA02 in Subjects with Non-Cystic Fibrosis Bronchiectasis and Chronic Pulmonary Pseudomonas Aeruginosa Infection (Tailwind)	Non-cystic Fibrosis Bronchiectasis; Pseudomonas Aeruginosa; Lung Infection	*P. aeruginosa*	2	Recruiting	Armata Pharmaceuticals, Inc. (Marina Del Rey, CA, USA)
Phage Therapy in Prosthetic Joint Infection Due to Staphylococcus Aureus Treated With DAIR. (PhagoDAIRI)	Infection of Total Hip Joint Prosthesis; Infection of Total Knee Joint Prosthesis	*S. aureus*	2	Recruiting	Pherecydes Pharma (Romainville, Paris)
Ph 1/2 Study Evaluating Safety and Tolerability of Inhaled AP-PA02 in Subjects with Chronic Pseudomonas Aeruginosa Lung Infections and Cystic Fibrosis (SWARM-Pa)	Cystic Fibrosis; Pseudomonas Aeruginosa; Pseudomonas; Lung Infection; Lung Infection Pseudomonal	*P. aeruginosa*	1/2	Recruiting	Armata Pharmaceuticals, Inc. (Marina Del Rey, CA, USA)
Safety and Efficacy of EcoActive on Intestinal Adherent Invasive E. Coli in Patients with Inactive Crohn’s Disease	Crohn’s Disease	*Adherent invasive E. coli* (*AIEC*)	1/2	Recruiting	Intralytix, Inc. (Columbia, MD, USA)
Study Evaluating Safety, Tolerability, and Efficacy of Intravenous AP-SA02 in Subjects with S. Aureus Bacteremia (diSArm)	SA Bacteremia (SAB)	*S. aureus*	1/2	Recruiting	Armata Pharmaceuticals, Inc. (Marina Del Rey, CA, USA)
A Study of LBP-EC01 in the Treatment of Acute Uncomplicated UTI Caused by Multi-drug-resistant E. Coli	Urinary Tract Infections	*E. coli*		Recruiting	Locus Biosciences (Morrisville, NC, USA)

* Denotes observational study for adverse events related to bacteriophage therapy.

## Data Availability

Not applicable.

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
