# Peer review of "Current Clinical Landscape and Global Potential of Bacteriophage Therapy"

_viruses, 2023, doi:10.3390/v15041020_

Round 1

Reviewer 1 Report

The following sentence:

"The country has the first ethically approved facility for phage treatment in Europe, the Hirszfeld Institute, located in the capital Warsaw. It is also accredited that the Institute has pioneered the compassionate use of therapeutic phage into modern medicine, contributing to the strengthening of knowledge about therapeutic phage [139]"

is repeated twice on the same page (in two different locations). If it wasn't enough the sentence contains a terrible mistake (also repeated twice). The Hirszfeld Institute is not located in the capital city of Warsaw (and never was).

There are also additional repeated parts across the manuscript. Table 2. is poorly formatted, hard to read although it contains little text. The following sentence "Consecutive administration of bacteriophages has been shown to result in gradually diminished efficacy, in part, due to the amplified production of neutralizing antibodies as part of the patient’s adaptive immune response" is only partially true. Papers describing that antibodies do not impact the efficacy of phage treatment in humans are widely available as well.

These days, reviews on phage therapy are numerous and mostly they don't bring anything new. There is no room for another one unless it is well written, original in its form, with self-made conclusions. This manuscript certainly does fit the above description.

I think this is enough to reject this manuscript as such quality is unacceptable. If none of the 9 authors had time to proofread this manuscript, I don't have time to read it further. Please respect the journal, the reviewer and try to focus on your work. Thank you.

Reviewer 2 Report

In this paper, the background, current situation, potential, and deficiency of phage therapy are described in detail.

However, several reviews have been published in the last three years, including in Viruses. Therefore, in order to provide new understanding and ideas on the basis of these review articles, this paper needs a major review.

First, reduce the description of antibiotics and resistant bacteria. This part of the story is why phage therapy is on the rise again, and it only takes a few numbers and a few words to explain it. I also recommend describing antibiotics first and then resistant bacteria.

Secondly, when I saw the title of the article, I thought that the article intended to review the clinical status and potential of phage therapy, but the article used a lot of space to introduce the basic knowledge of antibiotics, resistant bacteria, and phage. So I propose to reduce the description of the basic knowledge of phage, especially the classification and isolation section.

Third, the authors should provide more personal insights, such as how to deal with phage resistance in bacteria and how to deal with public concerns about phage therapy.

Reviewer 3 Report

The manuscript by Hitchcock et al. is an interesting and up-to-date review concerning the current status and potential of bacteriophage therapy with the inclusion of a broad background concerning the problem of antibiotic resistance in bacteria that justifies the development of phage therapy, the history of phage discovery and applications, and the phage isolation and analysis methodology. The authors provide extensive information concerning the cost of treating infections caused by antibiotic-resistant bacteria, the low rate of new antibiotics discovery and development, and the increasing number of failures of antibiotic therapies. In this view, even readers unfamiliar with bacteriophages and phage therapy should be convinced that phage therapy deserves development and introduction to practical use. A valuable part of the manuscript is the summary of currently ongoing and completed clinical trials on phage therapy efficacy and the list of bacteriophage collections and banks in different countries. The latter can serve as a source of practical information on where to look for phages against particular bacterial pathogens. In addition to citing numerous results of the successful use of phage therapy, the authors summarize phage therapy limitations and limitations that slow down the development of appropriate regulations allowing to introduce phage therapy into medical practice at a wide scale. The manuscript is well-written and interesting to read. I strongly recommend its publishing in Viruses. My minor comments are below:

L.83-84: Replace "antimicrobial-resistance bacteria have the capacity for horizontal transfer, spreading resistance across their" with "in most cases, bacterial antimicrobial-resistance determinants have the capacity for horizontal transfer, causing the spread of resistance within a"

L. 87: Transposable elements are not repetitive DNA sequences. Please correct.

L. 101: Replace "antibiotic therapies" with "antibiotics"

L. 323: Replace "it" with "their DNA"

L. 333- 336: The authors should mention somewhere here that the genetic load of lysogenic phages infecting bacterial pathogens commonly includes genes that increase the fitness of their bacterial hosts in the infected organism, such as, e.g., bacterial toxins or virulence determinants. Lysogenic conversion is an even bigger problem than transduction, a relatively rare event compared to lysogeny.

L. 350: The author should mention that the classification of bacteriophages based on morphological criteria was completely abandoned recently (see Turner et al., 2023; doi: 10.1007/s00705-022-05694-2.)

L. 359: Replace "Caudovirales" with "Caudoviricetes"

L. 368: bacterio-phages? Please correct

L. 388-398: Please verify and provide a reference.

L. 405-406: Replace "endolysin" with "endolysin-mediated"

L. 416 and elsewhere in the text: Replace "gram-negative" with "Gram-negative". Gram was the name of the staining method discoverer.

L. 456: Replace "water treatment plants" with "sewage treatment plants". Phages for therapy are most often isolated from communal or hospital sewage.

L. 457: Remove"which is"

L. 471: Explain what do you mean by TED talk. It is not obvious for all readers.

L. 497: Another earlier example is in Międzybrodzki et al., 2012, doi: 10.1016/B978-0-12-394438-2.00003-7. This should be also cited.

L.561: Standardization was a rule in Polish studies mentioned in Międzybrodzki et al., 2012, doi: 10.1016/B978-0-12-394438-2.00003-7. This should be mentioned.

L. 990: What kinds of labs do you mean? Do you mean research labs studying phage therapy. This should be specified. Otherwise, the title of Table 2 is misleading.

Author Response

Please see the attachment. Line numbering mentioned refers to the document with "Tracked changes accepted." 

Reviewer 4 Report

‘Current clinical landscape and potential of bacteriophage therapy’

The authors described form AMR, problems of AMR, economics, difficulties of antibiotic development and unmet needs of alternative antibiotics even though antimicrobial stewardship program in intro- contents. Non-professional person can understand the recent threats of AMR easily.

In bacteriophage overview, phage biology, classification & genomic diversity, host-spectrum determining RBP-receptor interaction, and some historical stories about phage findings were listed.

In phage therapy, from reigniting phage therapy in US in 2016 to high interests about phage therapy in western countries were introduced. Clinical trials that are conducting now are listed and introduced, which would be the most attracting contents in this manuscript.

In fact, for phage therapy to be successful, it is very necessary that legislative arrangements in each country help ensure that phage therapy is used in the right place at the right time. However, I do not foresee that phage therapy alone will be used for therapeutic purposes. I believe that sustained phage therapy with profit generation can be achieved when used in conjunction with antimicrobial agents. In many phage therapy-related reviews, the description of clinical trials in the present situation can be very helpful, but the scientific description of the combined treatment plan with phage and antibacterial agents is disappointing.

Round 2

Reviewer 1 Report

The manuscript has been significantly improved. Some minor amendments are still necessary though:

line 485: Please add "Hirszfeld" to the institute name, i.e. Hirszfeld Institute of Immunology and Experimental Therapy - this is an official English name. This is the same institution as in line 777.

line 518: clinicaltrials.gov - it would be a valuable addition to mention abut its European (no less important) counterpart - clinicaltrialsregister.eu and check what is listed there.

line 777: Hirszfeld (not Hirzfeld)

lines 1001-1002: "therapy's efficacy depends on adequate antibody induction" - again, this is only partially true. There are papers indicating that phage therapy in patients with low levels of antiphage antibodies failed contrary to patients with high level of antiphage antibodies (following phage application) in whom the treatment outcome was favorable. One might say that this makes no sense as the high level of antiphage antibodies should be responsible for phage neutralization. In fact, this could be easily explained - induction of antiphage antibodies speaks for the proper functioning of the immune system (often destroyed by the prolonged antibiotic treatment) which, indirectly, can help fight infection (despite the presence of phage-neutralizing antibodies). It must be explained further or simply please remove anything about antiphage antibodies. The belief that antiphage antibodies are the major obstacle in developing phage therapy is quite misleading these days.

Table 1 - please consider adding it using landscape orientation

Author Response

The manuscript has been significantly improved. Some minor amendments are still necessary though:

  1. Reviewer’s Comment: line 485: Please add "Hirszfeld" to the institute name, i.e. Hirszfeld Institute of Immunology and Experimental Therapy - this is an official English name. This is the same institution as in line 777.
    1. Authors’ Response: Thank you. This has now been fixed, with an updated abbreviation as well.
  2. Reviewer’s Comment: line 518: clinicaltrials.gov - it would be a valuable addition to mention abut its European (no less important) counterpart - clinicaltrialsregister.eu and check what is listed there.
    1. Authors’ Response: Thank you for sharing an additional resource to survey. There were two phage trials listed in clinicaltrialsregister.eu; the first trial is already included in Table 1 and the second is a completed trial from 2015. No other trials were identified to be added to Table 1. We have updated the language in line 526 to acknowledge the completed trial.
  3. Reviewer’s Comment: line 777: Hirszfeld (not Hirzfeld).
    1. Authors’ Response: Thank you. This has now been addressed.
  4. Reviewer’s Comment: lines 1001-1002: "therapy's efficacy depends on adequate antibody induction" - again, this is only partially true. There are papers indicating that phage therapy in patients with low levels of antiphage antibodies failed contrary to patients with high level of antiphage antibodies (following phage application) in whom the treatment outcome was favorable. One might say that this makes no sense as the high level of antiphage antibodies should be responsible for phage neutralization. In fact, this could be easily explained - induction of antiphage antibodies speaks for the proper functioning of the immune system (often destroyed by the prolonged antibiotic treatment) which, indirectly, can help fight infection (despite the presence of phage-neutralizing antibodies). It must be explained further or simply please remove anything about antiphage antibodies. The belief that antiphage antibodies are the major obstacle in developing phage therapy is quite misleading these days.
    1. Authors’ Response: We agree with the reviewer that the data available to examine the relationship between phage therapy and immunologic responses (notably antibody production) is conflicting and not well substantiated. We appreciate the theorized explanation for the antibody response that some studies have identified and agree with the reviewer that the discussion of antibodies should be removed. We have removed this language accordingly and will be vigilant for future studies that further explore the potential validity of this intriguing posited immunologic response.
  5. Reviewer’s Comment: Table 1 - please consider adding it using landscape orientation.
    1. Authors’ Response: This has been addressed.

Reviewer 2 Report

The MS meets the publication requirements of this journal.

Author Response

We thank the reviewer for their review and consideration of our manuscript.